# Synthetic molecular switches driven by DNA-modifying enzymes

Hong Kang[1,2], Yuexuan Yang [1] & Bryan Wei [1] ✉

Taking inspiration from natural systems, in which molecular switches are ubiquitous in the biochemistry regulatory network, we aim to design and construct synthetic molecular switches driven by DNA-modifying enzymes, such as DNA polymerase and nicking endonuclease. The enzymatic treatments on our synthetic DNA constructs controllably switch ON or OFF the sticky end cohesion and in turn cascade to the structural association or disassociation. Here we showcase the concept in multiple DNA nanostructure systems with robust assembly/disassembly performance. The switch mechanisms are first illustrated in minimalist systems with a few DNA strands. Then the ON/OFF switches are realized in complex DNA lattice and origami systems with designated morphological changes responsive to the specific enzymatic treatments.

Powered with superb programmability, DNA nanotechnology has emerged as a significant paradigm of rational design for nanoscale constructs[1,2]. Besides static nucleic acid structures with increasing complexity and delicacy under diversified architectures[3,4], dynamic constructs driven by strand displacement have also emerged as useful tools in computation and magnomechanics[5-7]. The capability to controllably switch ON and OFF nucleic acid devices[8,9] lays the foundation for the more advanced systems[10], including robotics[11-13], information processing[14], and computing[15-17]. More recently, dynamic constructs composed of complex DNA origami units have also been produced with similar design principles[18,19].

Natural molecular switches are common in biomolecular pathways in vivo, such as transcriptional regulation, cell signaling and cell division[20]. The most prominent example is the protein phosphorylation and dephosphorylation switch. A big collection of protein kinase and protein phosphatase is available to process a myriad of proteins at specific sites for sophisticated regulations[21]. In particular, divergent activities with opposite but complementary regulatory roles are presented for many kinases for more diversified regulation pathways[22,23]. Synthetic nucleic acid constructs can also be maneuvered by enzymes for controllable dynamics to be enabled (e.g., enzyme-driven DNA circuits[24,25], neural network[26], and DNA nanostructure reconfigurations[27,28]). Besides, systematic investigations into well-controlled reversible switches with forward and backward reactions are expected to reach the current trend in dynamic DNA

nanotechnology[29-32]. With a working interface between synthetic nucleic acid constructs and a full toolset of DNA/RNA processing proteins and especially enzymes available[33,34], such potential is yet to be unleashed to match structural DNA nanotechnology's rapid development.

One of the major challenges for the enzymatic treatments on DNA nanostructures is nonspecific interactions. For example, the off-target extension or cleavage can become more of an issue for DNA nanostructures with increasing complexity (MegaDalton to GigaDalton) to be treated with DNA polymerase or restriction enzymes, which could then lead to byproducts such as aggregates. It is desirable to add extra layers of guarantee to further reduce nonspecific reactions[35].

Here, we presented molecular switches for enzymatic treatments on diverse DNA nanostructures and special sequence features enabled the reliable reaction pathways. In our enzyme-mediated systems, the sticky ends of the synthetic DNA constructs were either exposed or blocked by a certain enzymatic treatment, which in turn resulted in the association or disassociation of structural partners to realize switches with forward and backward reactions. A minimalist system composed of a few strands was first designed to illustrate the reaction cycles to switch ON (assembled/coupled state) or OFF (disassembled/decoupled state) the sticky end cohesion. Then the concept was further extended to the controllable assembly and disassembly of complex lattices. To ensure the desired enzymatic pathways and minimize side reactions for our constructs from the scaffold-free molecular LEGO

[1]School of Life Sciences, Center for Synthetic and Systems Biology, Tsinghua University, 100084 Beijing, China. [2]Department of Biochemistry and Biophysics, University of Pennsylvania Perelman School of Medicine, Philadelphia, PA, USA. ✉e-mail: bw@tsinghua.edu.cn

method, we designed specific restriction sites for nicking endonuclease cleavage and three-letter coded segments for DNA polymerase extension alongside with the corresponding dual-thymine barriers (TpT barriers) to terminate extension. Since the M13 scaffold is not three-letter coded, dual-thymine tails (TpT tails) instead of TpT barriers were designed at the 3′ end of common staples in DNA origami constructs to prevent undesired extension from DNA polymerase. We also managed to synchronize the association and disassociation reactions to show a one-step 'cut' and 'paste' operation in complex DNA nanostructures.

## Results

### Basic designs for enzymatic-driven molecular switches

In our molecular systems, enzymes work as modulators to switch DNA nanostructures between ON and OFF states. To satisfy our switch design for the controllable catalysis of the forward and backward reactions, we selected DNA polymerase and nicking endonuclease with special characteristics. We screened DNA polymerases with strand displacing capability to dissociate paired sticky ends or open hairpins but without exonuclease activity of undesired digestion at either the 3′ or the 5′ end; we also assessed nicking endonucleases whose cut sites dwell outside the recognized region to create sticky ends of diversified sequences and with a stringent recognition sequence to minimize off-target cleavage. After testing several DNA polymerases and nicking endonucleases, we adopted $Bsu$ DNA polymerase, large fragment ($Bsu$ DNAP)[36], and Nt.AlwI nicking endonuclease[37] for our enzymatic implementation of two types of molecular switches (X and Y) with opposite reaction flow (Supplementary Fig. 1).

In type X switches (Fig. 1a), DNA polymerase drives the dissociation of the coupled structural partners (forward reaction X, FX), and nicking endonuclease drives the association of the decoupled partners (backward reaction X, BX). The initial coupled state (i.e., ON state) is designed to be two structural partners associated by sticky end segments $x/x_C*$ (Fig. 1a). Three-letter coding sequences of A, C, and G nucleotides (V nucleotides) for segment x and ones of C, G, and T nucleotides (B nucleotides) for segment $x_C*$, are specified for the enzymatic working sites. At the same time, TpT barriers (Supplementary Fig. 2a) are designed to stop the undesired extension. Upon $Bsu$ DNAP treatment for the forward reaction, extension takes place exclusively with segment x of V nucleotide as the template in the presence of dTTP, dCTP, and dGTP supply (step X1), and the newly synthesized ssDNA segment $x_E*$ displaces the original complementary sticky end $x_C*$ to disassemble the coupled partners (i.e., OFF state) in step X2. Extension with segment $x_C*$ as the template is disabled at the TpT barrier in the absence of dATP supply. Reversely, the newly extended segment $x_E*$ is nicked by Nt.AlwI (cut sequence 5′-GGATCNNNN↓N-3′) in reaction BX (step X3). Due to the transient breathing of nicked strand ↓$x_E*$ with segment x and the unfavored energetics (Supplementary Table 1), the hybridization competition from segment $x_C$ through a toehold-free strand displacement induces the dissociation of nicked strand ↓$x_E*$ (step X4). The original sticky end pair $x/x_C*$ reconnects two partners to a coupled complex (i.e., ON state).

In type Y switches, on the other hand (Fig. 1b), the same enzymes direct an opposite reaction cycle—nicking endonuclease drives the dissociation of the coupled structural partners (forward reaction Y, FY), and DNA polymerase drives the association of the decoupled partners (backward reaction Y, BY). The associated partners by sticky ends $y_C*/y$ can also be defined as the initial coupled state (i.e., ON state) (Supplementary Fig. 3). Upon Nt.AlwI treatment, the nicked strand ↓$y_E*$ is displaced by the hairpin stem folded back (step Y1) until a complete release. A kinetic disadvantage is present for ↓$y_E*$ when it becomes a separate strand. With an effective concentration of hairpin formation (≈30 mM) being much higher than the competing

hybridization of ↓$y_E*$/$y_H$ and y/$y_C*$ (≈250 nM), the kinetic favor drives the hairpin stem ($y_C*/y_H$) formation[38]. Consequently, the sticky end cohesion of $y_C*/y$ is canceled, leading the coupled state to the decoupled state (i.e., OFF state) in step Y2. Upon $Bsu$ DNAP treatment in subsequent step Y3, extension takes place exclusively with segment $y_H$ composed of V nucleotides as the template in the presence of dTTP, dCTP, and dGTP supply. Extension stops at the TpT barrier in the absence of dATP supply, thus precisely producing the extended segment $y_E*$ but not beyond. The extended segment $y_E*$ pushes open the hairpin stem and exposes the sticky end $y_C*$. As a consequence, the sticky end cohesion ($y_C*/y$) enables the coupling of the two structural partners back to the initial state in step Y4. At the same time, extension with segment y as the template is disabled at the TpT barrier in the absence of dATP supply.

Similar design principles of switches X and Y are applied throughout the study for an exclusive treatment dedicated to the working sites. Besides the working sites, other sticky ends are protected from nonspecific extension by TpT barriers. Since the custom sequence is not applicable to the M13 scaffold, TpT tails (Supplementary Fig. 2b) instead of TpT barriers are appended at the 3′ ends of common staples to serve the same protection purpose in DNA origami experiments. Exonuclease activity is absent for $Bsu$ DNAP, so the TpT tails remain at the 3′ ends of the staples upon treatment. Such protruding overhangs would not serve as proper initiation points of $Bsu$ DNAP extension, and hence the overall origami structural integrity would be preserved.

### Enzymatic implementation on DNA nanostructures

Next, we implemented these two molecular switches, X and Y in DNA nanostructures with specific arrangements of the working sites for switching between ON and OFF states. Molecular switch X (Fig. 2a) was first implemented in a minimalist system (Fig. 2b) as a complete complex X designed as two T-junctions (partners $X_I$ and $X_{II}$) connected by a single pair of sticky ends (16-bp). In reaction FX, upon $Bsu$ DNAP treatment at 37 °C for 2 h or overnight, the templated extension of $X_I$ displaced the original sticky end pair and, in turn, divided the complex X into $X_I$ and $X_{II}$ (from ON to OFF). Then in reaction BX, the newly extended segment from the last reaction was excised by the Nt.AlwI, and outcompeted by the sticky ends from $X_{II}$, resulting in the recombination of the complete X from the $X_I$ and $X_{II}$ with the displaced single strand as waste (from OFF to ON). The decoupled partners $X_I$ and $X_{II}$ were first treated by Nt.AlwI at 37 °C for 1 h and subsequently at 40 °C for 1 h or overnight for recombination.

Native polyacrylamide gel electrophoresis (PAGE) assays were used (Fig. 2b) to validate the efficacy of the enzymatic treatments and measure the corresponding yields. The yield of the FX reaction was measured as 90% in 2 h and that of the BX reaction as 12% for 24 h (Supplementary Figs. 4, 6). We then investigated a dual-pair binding system (Supplementary Fig. 7). The yield of FX dropped to 63%, and the yield of BX increased to 26% (Supplementary Figs. 8, 10). The yield increase of the backward reaction with an additional working site is consistent with the findings in our earlier study about the coordination of multiple binding pairs[17,39], neighboring sticky ends of the same construct created in the backward reactions enable the assembly in a coordinated manner in which the initial cohesion facilitates the subsequent ones due to a kinetic favor. Moreover, the ON/OFF switches of the dual-pair binding system were reversible (ON to OFF, back to ON, and then to OFF), with results shown in Supplementary Figs. 13 and 14. Since the forward and backward reactions proceed only with the respective enzymes, we presume that the removal of residue enzyme from the preceding reaction would be crucial for the dedicated enzyme to drive the successive reaction more efficiently, thus leading to more rounds of reversible reactions. Such a hypothesis was supported by the experiments with both enzymes supplied on purpose

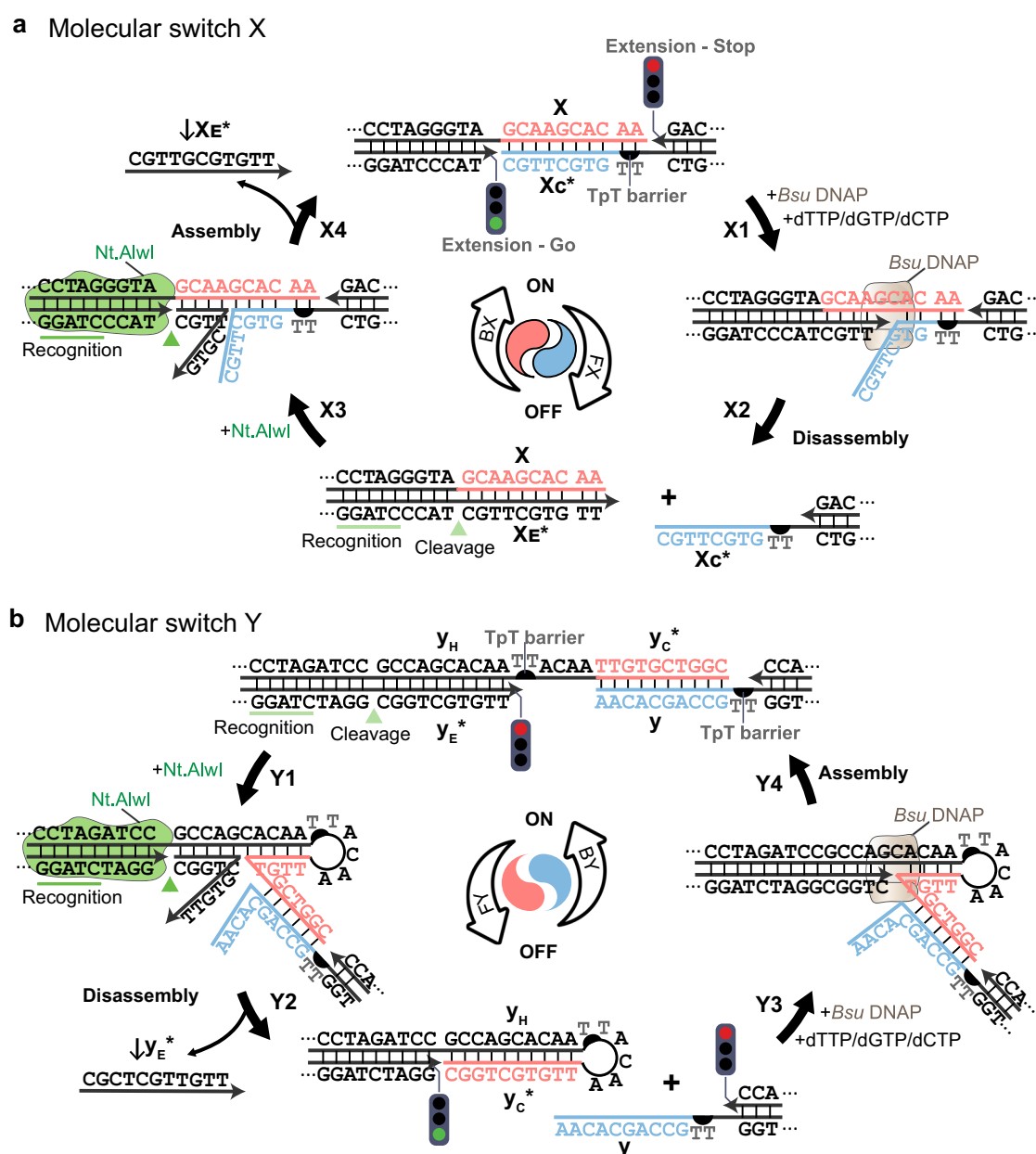

**Fig. 1 | Reaction schematics of two types of molecular switches. a** The reaction cycle of the type X switch comprises the FX reaction catalyzed by *Bsu* DNAP (a green light signal depicts the permissible start point of the extension, and a red light signal depicts the impermissible start point) and BX reaction catalyzed by Nt.AlwI (the recognition sequence underlined in green and cleavage site pointed by a green triangle). Extension of $x_E$* from *Bsu* DNAP displaced the segment $x_C$* (step X1), resulting in the disassembly of partners (step X2). The nicking of the newly extended segment $x_E$* by Nt.AlwI encourages coupling of the original binding partners x/$x_E$* (step X3), and the strand displacement leads the complex back to the coupled state (step X4). **b** The reaction cycle of the type Y switch comprises the FX reaction catalyzed by Nt.AlwI and BX reaction catalyzed by *Bsu* DNAP. The nicking of the segment $y_E$* by Nt.AlwI encourages hairpin formation that occupies the sticky end paring y and $y_C$* (step Y1), leading to the dissociation of the coupled complex (step Y2). Extension of $y_E$* from *Bsu* DNAP displaced the segment $y_C$* (step Y3), and the opening of the hairpin exposed the 3′ end segment $y_C$*, therefore resulting in the coupling of apart partners (step Y4). DNA sequence highlighted in blue is the complementary partner of the DNA sequence highlighted in red, which also serves as the template of the *Bsu* DNAP extension. Yin-Yang diagrams with (**a**) and without outlines (**b**) with the same blue-red color theme depict cyclic reactions of assembly/disassembly. Such a conceptual illustration is also adopted in Figs. 2–4.

(Supplementary Fig. 15). However, the thorough inactivation or removal of a chosen enzyme after its designated reaction was not carried out due to practical difficulties.

Furthermore, when compared to the toehold-mediated strand displacement system, apparent advantages were presented in our enzyme-mediated system. In FX reactions, although yields from the enzyme-mediated system (46%) and the toehold-mediated one (47%) were similar after a 2 h incubation (Supplementary Fig. 16a), the reaction rate of the enzyme-mediated system was much higher than that of the toehold-mediated system. As shown in the time course

results (Supplementary Fig. 16b), the enzymatic reaction reached an equilibrium in less than 3 min (reaction rate at 3 min⁻¹), while it took more than 90 min to reach a similar level of equilibrium in control experiments of toehold-mediated strand displacement (reaction rate at 0.03 min⁻¹). In BX reactions (Supplementary Fig. 17), the assembly yield of the enzyme-mediated system was measured at about 29%, but no obvious assembled product was generated in our control of the toehold-mediated system.

Type X switch was then implemented in a 5 × 5 lattice with 10-nt sticky ends for the 4-arm junction inter-motif complementation

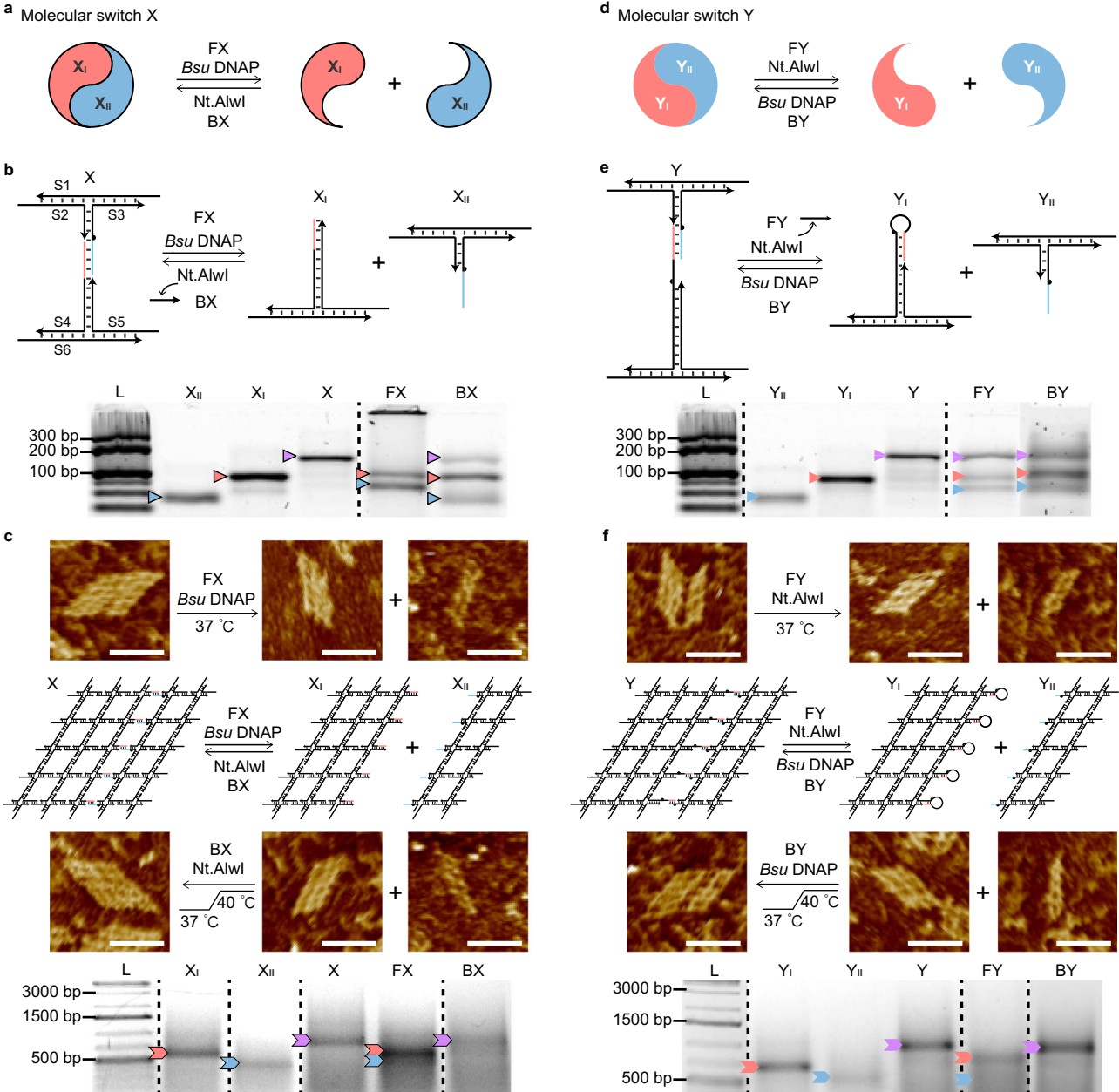

**Fig. 2 | Implementation of molecular switches on DNA nanostructures.**
**a** Conceptual diagrams of type X ON−OFF switches. **b** Minimalist system of type X switch. Schematic diagrams and the corresponding PAGE results are shown. **c** 5 × 5 lattice system (5′ end 10-nt sticky ends designed for the 4-arm junction inter-motif complementation) of type X switch. **d**–**f** Conceptual diagrams of type Y ON−OFF switches (**d**), diagrams and results in the minimalist system (**e**), and diagrams and results in 5 × 5 lattice system (**f**). Yields of reactions FX and BX are in (**b**) 90 ± 1% and 12 ± 1%, respectively; yields of reactions FY and BY in (**e**) are 42 ± 2% and 19 ± 5%, respectively. Triplicated experiments in Supplementary Figs. 4−6. A significant yield increase was presented in the assembly of the dual-pair binding system (triplicated experiments in Supplementary Figs. 8−11). The X and Y ON/OFF switches were operated reversibly in Supplementary Figs. 13 and 14. Reaction diagrams in **c** and **f** are sandwiched by the AFM results of forward and backward reactions. Scale bars: 50 nm. The corresponding PAGE results are also shown. The triangles/chevrons in red, blue, and purple point to the product bands of $X_I$ ($Y_I$), $X_{II}$ ($Y_{II}$) and X (Y), respectively.

(Fig. 2c, Supplementary Figs. 18a, 24)[40]. Five type X working sites were designed to bridge the third and fourth columns from left to right (shown in red and blue). After *Bsu* DNAP treatment at 37 °C for 4 h, the complete 5 × 5 lattice (X) was divided into a 3 × 5 lattice $X_I$ and a 2 × 5 lattice $X_{II}$ (as major bands shown in gel results: Fig. 2c, lane FX, Supplementary Fig. 20). Then the two lattices ($X_I$ and $X_{II}$) recoupled to the complete one with Nt.AlwI treatment (as a dominant band shown in gel results: Fig. 2c, lane BX, Supplementary Fig. 21). Only product bands but not leftover substrates are present in the gel results (a yield of 32% for FX and a yield of 16% for BX). Diffused target bands have presumably resulted from the non-specific DNA−enzyme interaction.

When comparing the high yields with the ones from the simpler constructs with fewer working sites (Fig. 2b), consistent with the yield improvement in minimalist constructs with an additional working site (Supplementary Fig. 12), the results of constructs with multiple working sites indicate that the cooperative reactions of multiple working sites along the lattice edges can drive the overall switch reactions synergistically into completion. The product bands were excised out from the agarose gel, and the eluted samples of purification were subjected to atomic force microscopy (AFM) imaging (Supplementary Figs. 26, 27). The morphologies of the products from target bands were in good agreement with our expectations (Fig. 2c).

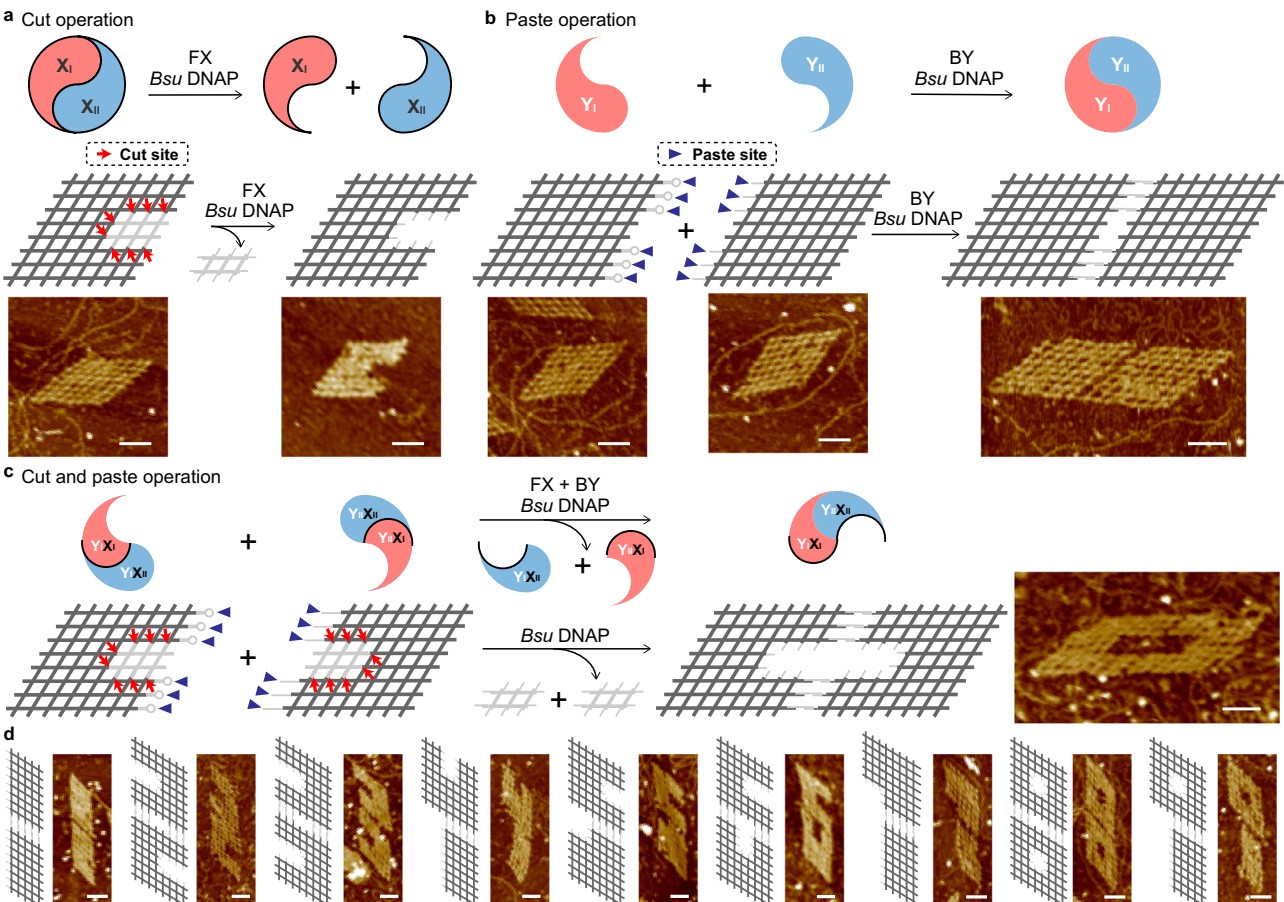

**Fig. 3 | One-step cut and paste operation on 8×8 lattice system. a** Schematics and results of cut operation (i.e., FX reaction). The AFM results before and after enzymatic treatment match the design. **b** Schematics and results of paste operation (i.e., BY reaction). The AFM results before and after enzymatic treatment match the design. **c** Schematics and results of the composite cut (FX reaction) and paste operation (BY reaction). Combining the cut-and-paste sites from (**a**) and (**b**), a 0-shaped lattice can be created by *Bsu* DNAP. AFM results after the enzymatic treatment matches the design. **d** Schematic and results of cut and paste products of digit-shaped lattices from 1 to 9. Reaction strand diagrams and the corresponding AFM results after enzymatic treatment are shown side-by-side Scale bars: 50 nm.

In parallel, the type Y switch (Fig. 2d) was also implemented in systems based on T-junctions and 5 × 5 lattices. Under this framework, the working site was designed for the single-pair binding system (Fig. 2e). The yield of reaction FY at 37 °C overnight was 42% (Supplementary Fig. 6), and the yield of reaction BY at 37 °C for 1 h and then 40 °C for 1 h was 19% (Supplementary Fig. 5). Consistent with the trend in the dual-pair system of type X, the yield of dual-pair system FY dropped to 23% (Supplementary Fig. 11) and that of BY increased to 60% (Supplementary Fig. 9). Again, cooperativity accounted for the BY yield increase presumably (Supplementary Fig. 12).

In the 5 × 5 lattice system of type Y (Fig. 2f, Supplementary Fig. 25), the Nt.AlwI nicking endonuclease treatment resulted in the dissociation of the complete 5 × 5 lattice Y into a 3 × 5 $Y_I$ and a 2 × 5 $Y_{II}$ at 37 °C for 4 h reaction FY, with a yield of 17% as major bands shown in gel results (Fig. 2f, lane FY, Supplementary Fig. 22). Then *Bsu* DNAP reopened the hairpin loops for the two lattices $Y_I$ and $Y_{II}$ to recouple into the complete lattice Y in reaction BY, with a yield of 18% as a dominant band shown in gel results (Fig. 2f, lane BY, Supplementary Fig. 23). Similar to the aforementioned results about type X switches, we attributed the high yields to the cooperativity of the clustered working sites. The gel-purified products were subjected to AFM, demonstrating the effective switching between ON and OFF states (Fig. 2f).

## One-step cut and paste operation in DNA lattice structure

To demonstrate the divergent activity of the same enzyme in two independent molecular switches, we proceeded to execute association and dissociation operations simultaneously within one round of reaction with a single species of the enzyme (e.g., *Bsu* DNAP). We designed X working sites (i.e., 'cut' sites) for internal parts to be carved out and external Y working sites (i.e., 'paste' sites) for two or more structure units to get combined. A composite 'cut' and 'paste' operation was executed with DNA polymerase treatment at the same time in one pot. It is also feasible to design a composite operation based on nicking endonuclease (Supplementary Fig. 28). Due to the higher reaction yields (FX > FY, BY > BX, Suppl. Fig. 12), we focused on DNA polymerase for the composite operation in this work.

To start with, we sought to operate cut and paste individually on the 8×8 lattices with 14-bp sticky ends for the inter-motif complementation (Supplementary Figs. 18b, 29, 38). In the cut operation, eight-cut sites of reaction FX were arranged inside for the *Bsu* DNAP cutting, highlighted by the red arrows (Fig. 3a). Upon treatment of *Bsu* DNAP at 37 °C for 4 h, the templated extension displaced the original internal sticky end pairing and dissociated the 8 × 8 lattice into a large C-shaped lattice and a small 2 × 3 lattice (as waste). The agarose gel assays were used to analyze the results (Supplementary Fig. 30), and the C-shaped lattice was presented under AFM (Fig. 3a, Supplementary Fig. 39).

As for the paste operation, external sextuple-pair binding sites of reaction BY as paste sites of the two 8 × 8 lattices were designed at the chosen sides highlighted by the purple triangles (Fig. 3b), respectively. Upon *Bsu* DNAP treatment at 37 °C for 4 h, the hairpins at one side were opened to become the sticky ends, in turn, resulted in the coupling of

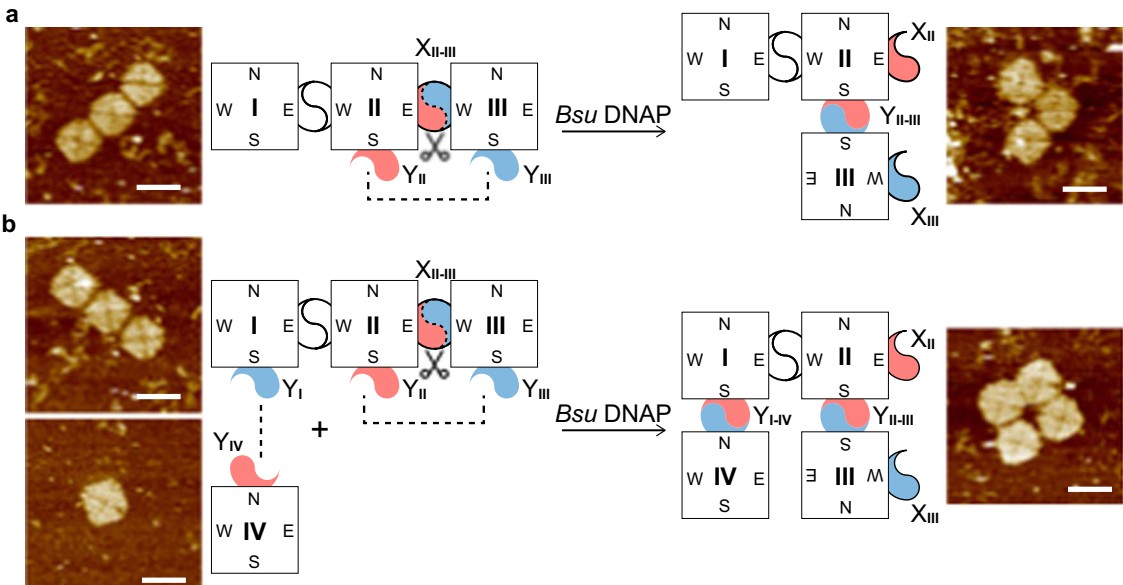

**Fig. 4 | Complex one-step cut and paste operations in origami square system.**
**a** Composite cut and paste operation from I-shape trimer to L-shape trimer. The cut sites were designed at the east side of unit II and the west side of unit III, and the paste sites at the south edge of unit II and the south edge of III. After a one-pot cut-and-paste operation by *Bsu* DNAP, the L-shaped trimer can be specifically created. **b** Composite operation from monomer and I-shape trimer to O-shape tetramer.

Based on (**a**), an extra unit IV with paste sites at the north attached to the south of unit I to form an O-shape tetramer. The scissors depict cut operations to split units II and III. The dash lines depict the paste operations to connect units II and III. AFM results are shown side by side with the substrate and product of the reaction diagram, respectively. Scale bars: 100 nm.

two 8 × 8 lattices by sticky end cohesion. The agarose gel (Supplementary Fig. 31) and AFM (Fig. 3d, Supplementary Fig. 40) results agreed well with the designated outcome of the combined 8 × 16 lattice.

After the successful implementation of individual operations, we continued on for a composite operation with both cut sites and paste sites included (Fig. 3c). Specifically, eight-cut sites and six-paste sites were designed for each of the two 8 × 8 lattices (Fig. 3c). Upon *Bsu* DNAP treatment at 37 °C for 4 h, the two lattices were carved into C-shaped lattices (with the opening facing left and right, respectively) at the cut sites and coupled at the paste sites. This can be abstracted as $Y_I$ decoupled into $Y_IX_I$ and $Y_IX_{II}$ and $Y_{II}$ into $Y_{II}X_I$ and $Y_{II}X_I$, while the $Y_IX_I$ and $Y_IX_{II}$ coupled at the paste sites with $Y_IX_{II}$ and $Y_{II}X_I$ as wastes. Collectively, a 0-shaped complex resulted from a composite cut (carving of two C-shaped lattices) and paste (coupling of the two carved lattices) operation. The resulting lattice was verified by agarose gel assay (Supplementary Fig. 32) and AFM (Fig. 3c, Supplementary Fig. 21). With transferring cut sites inside of the lattice, different shapes to be carved out of the 8 × 8 lattices and other shapes of digits (digit numbers 1–9) were produced following similar design principles and operation (Fig. 3d). The corresponding rearranged digit lattices were analyzed by agarose gels (Supplementary Figs. 32–37) and present under AFM (Supplementary Figs. 41–50) Exposure of single-stranded overhangs over a certain threshold was a known contributor of compromised self-assembly yield, and such a phenomenon was reported in early studies of DNA nanostructures without enzymatic treatment[41]. A certain cut operation resulted in the creation of many single-stranded overhangs, which collectively led to random aggregation due to nonspecific base pairing. The formation yield variation of different shapes was substantial, and it related more to single-strand exposure than enzymatic treatment. Therefore, yield quantification was not provided in this group of constructs.

We anticipate the composite cut-and-paste operation to be applied at a more intricate level. As shown in one of the several examples from our schematic presentation (Supplementary Fig. 51), lattice '1' and lattice '2' could be combined into lattice '3' (as the sum of

'1' and '2'). Furthermore, one could also use Cas9 to substitute nicking endonuclease to achieve a more arbitrary cleavage without relying on predefined working sites.

## One-step cut and paste operation in DNA origami structure
Carefully designed DNA origami squares[42] were then used to demonstrate the one-step cut-and-paste operations with more structural units. Since custom three-letter coding was not applicable for the M13 origami scaffold (Supplementary Fig. 52), protective TpT tails (Supplementary Fig. 2b) were introduced to ensure the exclusive treatment without undesired modifications on common staples. Specifically, a TpT tail was appended at the 3′ end of each common staple. Such an unpaired overhang remained undigested from the enzymatic treatment because of the lack of exonuclease activity for *Bsu* DNAP. As a consequence, the 3′ TpT tail was not a proper starting point for *Bsu* DNAP to initiate segment extension, thus preventing the undesired strand displacing to maintain structural integrity (Supplementary Fig. 53). On the other hand, 12-bp sticky ends of user-specified sequences were adopted to combine origami units, and typical TpT barriers were designed for these strands to avoid undesired extension.

In the first example (Fig. 4a), an I-shape trimer of three units to be initially combined in tandem was designed to be converted into an L-shape trimer upon *Bsu* DNAP treatment. The cut-and-paste sites X and Y were designed at the chosen edge sides of specific units for the designated rearrangement. Specifically, five cut sites were assigned to the east side of unit II (as $X_{II}$) and the west side of unit III (as $X_{III}$); five paste sites were assigned to the south side of unit II (as $Y_{II}$) and the south side of unit III (as $Y_{III}$). Upon treatment of *Bsu* DNAP at 37 °C for 1 h and at 40 °C for an additional 4 h, unit III detached from the east side of unit II and then reattached to the south side of unit II. As a consequence, the I-shape trimer was successfully converted into an L-shape trimer (Fig. 4b). Agarose gel electrophoresis assays (Supplementary Fig. 53) were used to verify the structures. The experimental bands up-shifted and then controlled bands after being treated by the enzyme. AFM results showed the distinct shapes (Fig. 4b, Supplementary Figs. 54–56).

Similarly, in the second example, when one more origami unit IV was added, with the paste site $Y_I$ appended on the south side of unit I and $Y_{IV}$ appended on the north side of unit IV, an O-shaped tetramer was converted from an I-shaped trimer (I–II–III) and a monomer (IV) after enzymatic treatment. As a control, when the cut sites were not available for units II and III, the I-shape trimer remained unchanged while unit IV attached to form an L-shape tetramer after enzymatic treatment (Supplementary Fig. 57). The desired operation was also verified by agarose gel electrophoresis (Supplementary Figs. 58, 59) and AFM imaging (Fig. 4b, Supplementary Figs. 60, 61).

We noted a lower yield when compared to the results based on 5 × 5 lattices. As suggested by monomer and dimer intermediates shown in the gel results (Supplementary Fig. 53), such a compromised yield is presumably due to non-optimal conditions for the one-pot reaction of the extension by *Bsu* DNAP and the assembly from 12-nt sticky end cohesion. Adjustment of buffer condition and/or sticky end length might be useful for yield improvement.

## Discussion

In conclusion, our molecular systems for enzymes to controllably switch ON and OFF sticky end cohesion were presented. Well-defined assembly/disassembly behaviors of synthetic DNA nanostructures were shown as responsive to specific enzymatic treatments. We believe this rational design scheme of enzymatic treatments as implementation handles in DNA nanostructures is a meaningful addition to the platform of enzyme-free strand displacement. According to the side-by-side comparison with the control system based on toehold-mediated strand displacement, the reaction yield and/or rate of our enzyme-mediated system was superior (Supplementary Fig. 16), not to mention the simpler components and independence of stoichiometry.

The direct ON or OFF signal of our molecular switches was either exposure or blocking of sticky ends and the corresponding downstream reaction chain of association or disassociation of structural partners is not in an optimized efficiency. Side reactions such as random aggregation could disrupt the reaction chains. In our synthetic construct design, functional sequences already serve as checkpoints to ensure the desired reaction pathways while reducing spurious reactions. It would be appealing to recruit additional components (functional DNA segments and enzymes) for a more efficient and specific switch (e.g., less off-target reactions). An inactivation of specific enzymatic activity or a total removal of a certain enzyme at a precise time point might also be useful to prevent nonspecific reactions.

In natural biomolecule systems, ON and OFF signals are of many kinds (e.g., phosphorylation, methylation, and ubiquitination) that center around the intricate regulation network of life activities, and they are coupled to diverse downstream cascades, often with superior efficiency. Moreover, positive and negative regulators played a crucial role in the biomolecular constructions of various types of cells (e.g., neurons and epithelial cells) for spatiotemporal regulation of partial destruction of existing supramolecular constructs and gradual build-up of new ones by simple and repetitive structural units. For example, the microtubules in existing cytoskeleton structures, such as those in primary dendrite or cell body, can be trimmed into short fragments (e.g., by negative regulators), which are simultaneously stabilized and then reorganized (e.g., by positive regulators) to form new microtubule arrays in the formation of new sub-cellular compartments, (e.g., neuronal branches, synaptic boutons and cilia[43,44]). To take inspiration from Nature, it is therefore desirable to design more efficient reaction cascades with multi-layer regulatory components.

As shown in our systems of this study and earlier ones in the literature, many enzymes can naturally fit in the enzymatic frameworks, and common design elements in DNA dynamics (e.g., hairpins) are also perfectly compatible. With the enriched space of design and engineering, one can imagine controlled synthetic systems with switches regulated by diverse elements to attain designer functionalities.

## Methods

### Structural design and DNA sequences

Strands used in minimalist systems and 2D single-stranded tile structures (5 × 5 lattice and 8 × 8 lattice) were designed and generated through Uniquimer[45] software (version 1.0). DNA strands used in simple systems were ultra-PAGE synthesized by Sangon Biotech. DNA strands used in lattice structures were synthesized by Integrated DNA Technology, Inc. The core strands of the 2D origami square were derived from the design by Qian and coworkers[42]. Yet the edge strands were designed and generated through Uniquimer software. All sequences used in this work are provided in Supplementary Dataset 1.

### Structural construction

In the assembly of the minimalist system, strands were mixed in a roughly equimolar concentration of 500 nM for dimer and 750 nM for monomers in 1× TAE buffer (40 mM Tris pH 8.0, 20 mM acetic acid, and 1 mM EDTA) supplemented with 15 mM $MgCl_2$. The samples were annealed in a thermocycler (90 °C for 3 min and cooling from 90 to 25 °C over a period of 1 h) before reacting with different enzymes. To assemble SST squares (5 × 5 lattice and 8 × 8 lattice), the component DNA strands were mixed in a roughly equimolar concentration of 400 or 800 nM in 1× TAE buffer (40 mM Tris pH 8.0, 20 mM acetic acid, and 1 mM EDTA) supplemented with 40 mM $MgCl_2$. The sample was annealed in a thermocycler (94 °C for 5 min, and cooling from 90 to 25 °C for 72 h) before enzymatic reactions and native agarose gel electrophoresis. To assemble the origami square, 10 nM scaffold M13mp18, 50 nM core staples, and 100 nM connection staples were mixed in 1× TE buffer (10 mM Tris pH 7.9, 2 mM EDTA) supplemented with 12.5 mM $MgCl_2$. For each structural assembly and intermolecular interaction, there are more than three independent experiments conducted.

### Enzymatic reaction

In each reaction, 1×*Bsu* DNA polymerase (NEB) and 1×Nt.AlwI were added to the structures with 1× corresponding buffer at 37 °C for disassembly, 42 °C for assembly in a simple system, and 40 °C for other structures.

### Agarose gel electrophoresis and structural purification

1%, 2%, and 8% agarose gel were prepared in 0.5× TBE buffer (45 mM Tris pH 8.0, 45 mM boric acid, 1 mM EDTA) supplemented with 10 mM $MgCl_2$ and pre-stained with SYBR Safe (Thermo Scientific). The annealed products were subjected to native agarose gel electrophoresis at 90 V in an ice-water bath. Then the target gel bands were excised by the blade, carefully crushed using the flat end of a plastic pestle in a Freeze'N Squeeze column (Bio-Rad), and then directly subjected to centrifugation at 106×*g* for 2 min at 4 °C or room temperature. Samples centrifuged through the column were collected by SynGene_GeneTools software (version 4.03.05.0).

### AFM imaging

The SPM Multimode with Digital Instruments Nanoscope V controller (Vecco) was used to collect AFM images of purified DNA nanostructures, and the NANOSCOPE ANALYSIS (version 1.50; Bruker Corp.) was used to analyze the height of structures. A 50 µL drop of solution for structure assembly and a 2–3 µL droplet of the sample were applied to a freshly cleaved mica surface and left for ~2 min incubation. The images were captured under liquid tapping mode, with C-type triangular tips (resonant frequency, $f_0 = 40$–75 kHz; spring constant, $k = 0.24$ N m$^{-1}$) from the SNL-10 silicon nitride cantilever chip (Bruker Corporation). For AFM imaging of each structure, independent triplicated experiments were conducted.

## Yields quantification by agarose gel electrophoresis

In native agarose gel electrophoresis, the intensity of target bands was used to estimate the reaction yields. To measure the yields of structure formations and enzyme reactions, the ratio between the fluorescent intensity of a target band and the intensity sum of the entire lane was taken to represent the yields. Software ImageQuantTL (GE Healthcare, version 10.2) was used to calculate the fluorescent intensity.

## Reporting summary

Further information on research design is available in the Nature Portfolio Reporting Summary linked to this article.

## Data availability

The data that support the findings of this study are available from the corresponding author upon request.

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

## Acknowledgements

We thank Xin Liang and Qing-Shan Jia for the discussions. This work is supported by the National Key R&D Program of China (grant No. 2021YFF1200200), Tsinghua University Spring Breeze Fund, and Tsinghua University-Peking University Joint Center for Life Sciences (to B.W.).

## Author contributions

H.K. designed the experimental systems, performed experiments, analyzed the data, and wrote the manuscript; Y.Y. performed experiments about the yield analyses of the minimalist systems and edited the manuscript. B.W. conceived and supervised the project, analyzed the data, and wrote the manuscript.

## Competing interests

The authors declare no competing interests.
