## [Peer Review File · Nature Communications]

Synthetic molecular switches driven by DNA-modifying enzymesREVIEWER COMMENTS

Reviewer #1 (Remarks to the Author):

The key result of this manuscript is experimental demonstration of an enzyme-powered DNA logic on/off switch to drive rearrangement of DNA array and DNA origami nanostructure components. The enzyme-drive switch is demonstrated first as an isolated complex, where a DNA polymerase drives one reaction and a nicking enzyme the backwards reaction. There are 2 different versions (X,Y) with a slightly different geometry, such that the same enzymes drive the switch in different directions. For X circuit polymerase drives on to off transition and nicking enzyme drives off to on, and for Y circuit vice versa. This switch is then adapted for use in the connectors of finite size DNA tile shapes (5x5 array) and in links between DNA origami squares.

There exist a number of other enzyme driven DNA logic systems in the literature, the novelty here is the adaptation for use in the context of larger DNA nanostructures. The key challenge to this is that many DNA polymerases show spurious nucleation at non-target sequences, similarly nicking enzymes can show low but not zero off-target cleavage of DNA. In the context of a large (10,000 base pair) DNA origami this can lead to lots of off target effects, leading to rapid degradation of the DNA nanostructures. The main advance of the author's work is to demonstrate a system that can have both enzyme driven logic and DNA nanostructures, by careful control of 3 nucleotide DNA sequences in the 5x5 array and choice of a specific DNA polymerase that is blocked by the TpT barrier sequence, which is then added to 3' staple ends of the DNA origami. This is a interesting and important advance in the field of switchable DNA origami structures, with potential to open up lots of future directions. The on/off switch is then used as a 'cut' and 'paste' function to build up a range of different shapes, such numbers 1-9. These results are supported by some nice AFM images of the resulting structures.

The main weakness of the work is the claims of very high yields for some of the cut and paste transitions, eg. page 7 'The corresponding yields ranged from 20% to 60%'
The data in supplementary information does not support these yield numbers. For example, SI figure 24 the 0 and 5-shaped lattices estimates yields as 30% (lanes 2,3) and 61% (lanes 5,6). The method of gel yield is to calculate the product band intensity over the total lane intensity, but in almost all gels following enzymatic reactions there is a very large and high intensity high-molecular weight smear in the gel and also clearly a lot of aggregates in the well. This includes gels in S Figs 22,24,25,26,27,28. Both the gel smear of higher molecular weight aggregates overlapping the product band and material remaining in the well will lead to an overestimate of the yield using the author's methods. Indeed, in S Fig 28 the claim is of yield 20%, when instead very obviously almost the entire sample is aggregated to the point it remains in the well. The yield claims are also not backed up by the microscopy data, which show only 1-2 correctly formed structures for each sample and a majority of DNA is in incorrectly assembled structures or large aggregates. The authors need to address these unsupported claims of high yields before the paper can be published.

The following issues also need to be addressed:

1. introduction - more clearly identify what is lacking in the current state-of-the-art
2. Add citations to the following key papers in enzyme-driven DNA logic circuits that involve polymerase/degradation pairs of enzymes:
Okumura, S., Gines, G., Lobato-Dauzier, N. et al. Nonlinear decision-making with enzymatic neural networks. *Nature* 610, 496–501 (2022). <https://doi.org/10.1038/s41586-022-05218-7>
Schaffter, S.W., Schulman, R. Building in vitro transcriptional regulatory networks by successively integrating multiple functional circuit modules. *Nat. Chem.* 11, 829–838 (2019). <https://doi-org.ezproxy.library.sydney.edu.au/10.1038/s41557-019-0292-z>

3. Include references for other work that has used different methods to prevent off-target interactions of polymerases with DNA origami

Extrusion of RNA from a DNA-Origami-Based Nanofactory. Jaeseung Hahn, Leo Y.T. Chou, Rasmus S. Sørensen, Richard M. Guerra, and William M. Shih ACS Nano 2020 14 (2), 1550-1559 DOI: 10.1021/acsnano.9b06466

4. Page 2, design of switches - 'After testing several DNA polymerases and nicking endonucleases...'
- add more details on why Bsu DNAP and Nt.AlwI were selected, what other enzymes were tested and principles of selection

5. Page 2: 'dual-thymine barriers (TpT barriers) are designed to stop the undesired extension...'
- provide more detail on mechanism for this sequence preventing further extension, this is a key feature of the later approach to DNA origami and should be emphasised with more detail earlier in the text

6. page 3 'complementation' doesn't make sense here, should be 'hybridization'

7. page 4 'system can be cycled three rounds'

- a full cycle or round implies on-off-on, here the results are for on-off-on, so at most 1.5 cycles
- however, the gel data for this claim in the supporting figures SI Fig 10/11 is not clear at all. Is step 1 in these gels an enzyme-driven step? Or is it just assembly of the starting material? For step 3 (lanes 3/4 in SFig 10) there is no reassembled 'X' product band visible in this gel at all, just a slightly darker smear. Similarly for step 3 in (lanes 6 in SFig 11) for the Y-design, the product band indicated by the purple arrow is barely there and not very convincing.

8. Generally, annotation of gels in paper and SI should more clearly indicate which samples are a result of enzyme addition, and which are DNA complexes assembled as starting materials

9. Most results show that the 'cut' operation function works much better than the 'paste' function, this needs to be addressed in the discussion

10. page 4 'cooperative reactions of multiple working sites along the lattice edges can drive the overall switch reactions synergistically into completion'

This explanation makes sense for the 'paste' function, as having more possible sites to nucleate reassembly of the components would likely improve the yield. It is not clear how cooperativity could help with disassembly in the 'cut' function as well? It would be more likely to expect cooperativity to reduce the yield of 'cut' as many more bonds have to break to remove the small unit

Reviewer #2 (Remarks to the Author):

In this paper, the authors demonstrate the use of a nicking enzyme and polymerase action for the reconfiguration of DNA nanostructures. They first present a concept in which two DNA complexes can associate with each other and subsequently are made to dissociate. Essentially, association leads to the formation of a recognition site for a nicking enzyme. This nicking site is used as a start site for a strand-displacing DNA polymerase. Polymerization is controlled by the introduction of TT sites and the omission of dATP in the nucleotide mix.

The authors then demonstrate that a DNA lattice can be cut at predefined sites, and two lattices can be rejoined. They also show that specific shapes can be cut out from such a lattice, and that

structures consisting of several connected DNA origami structures can be reconfigured.

Generally, using such enzyme-driven strand modifying operations is a very interesting concept, which is potentially useful for the realization of dynamically reconfigurable nanostructures from DNA. The overall concept is somewhat reminiscent of the "DNA toolbox" approach by Rondelez et al., which also relies on nicking enzymes and DNA polymerase strand displacement for the realization of dynamic systems.

However, it appears that in this work, the authors do not fully realize the potential of the approach, and the progress made is not as significant as one might expect. Specifically, it is not clear how programmable this approach is in the end, how quickly the processes occur (compared to, e.g., ordinary strand displacement), the robustness of the procedures/their yield, and ultimately, whether there is any significant advantage over existing or alternative methods. These concerns should be addressed in a revised version of the manuscript.

Specific points:

- What is the "Yin and Yang theory"? This reviewer believes that Yin and Yang are not scientific concepts and are not well-defined. This should be phrased more carefully, e.g., "similar to Yin and Yang, the reactions complement each other, etc...". It is notable that the term "Yin and Yang" only appears in the title of the cited reference 23, likely because it sounds appealing, but not more.
- In the reaction cycle in Fig. 1b, it is unclear to the reviewer whether switch Y would predominantly be present in the extended form (open hairpin) on top or if it would rapidly interconvert between the hairpin form (on the right) and other intermediates. Perhaps the hairpin is even more thermodynamically stable than the extended form?
- The reviewer does not understand why one of the states is labeled "ON" and the other "OFF".
- The overall yields of the reactions are not particularly impressive. Given this, what is the advantage of a procedure that requires the addition of enzymes, resulting only in partial reconfiguration? Multiple cycles will further reduce the overall yield. Maybe one can highlight and discuss the differences to other approaches for reconfigurability, and why this one is superior. If it is not, what has to be developed?
- While the concept in Fig. 1 promises the ability to switch between two different designs, this is only demonstrated once with DNA nanostructures (Fig. 2c and 2f). Figures 3 and 4, on the other hand, are solely based on the polymerase-driven FX and BY (which might be more efficient?). This suggests that the potential strengths of the concept aren't fully realized in its current form.
- Cutting and rejoining reactions are not freely programmable. Essentially, the break/cutting points must be designed into the devices. Also there are sequence constraints due to the use of a 3-letter code. Developing more general reconfigurability would be intriguing.
- In Fig. 4, the authors use the term 'reconfiguration'. It should be clarified whether the structures individually undergo internal reconfiguration (possibly by linking the substructures together) or if they all merely disassemble and reassemble in a different form.

Reviewer #3 (Remarks to the Author):

Nice work! Kang and Wei have developed a DNA switch system driven by a combination of DNA restriction nicking enzyme and DNA polymerase with strand displacement capability. The two enzymes

work in the opposite directions: one enzyme works to associate two DNA molecules and the other works to dissociate the two DNA molecules. The functions of the two enzymes in terms of association-dissociation depends on the specific design; but in either case, they are specific and controllable. The authors have applied this strategy to multiple systems of DNA nanostructures. Each system has been concretely demonstrated by native gel electrophoresis and atomic force microscopy. Excellent idea and concrete experimental demonstration! This work is a great addition to the tool box of regulating DNA self-assembly. This referee would strongly recommend for publication.

REVIEWER COMMENTS

Reviewer #1 (Remarks to the Author):

The key result of this manuscript is experimental demonstration of an enzyme-powered DNA logic on/off switch to drive rearrangement of DNA array and DNA origami nanostructure components. The enzyme-drive switch is demonstrated first as an isolated complex, where a DNA polymerase drives one reaction and a nicking enzyme the backwards reaction. There are 2 different versions (X,Y) with a slightly different geometry, such that the same enzymes drive the switch in different directions. For X circuit polymerase drives on to off transition and nicking enzyme drives off to on, and for Y circuit vice versa. This switch is then adapted for use in the connectors of finite size DNA tile shapes (5x5 array) and in links between DNA origami squares.

There exist a number of other enzyme driven DNA logic systems in the literature, the novelty here is the adaptation for use in the context of larger DNA nanostructures. The key challenge to this is that many DNA polymerases show spurious nucleation at non-target sequences, similarly nicking enzymes can show low but not zero off-target cleavage of DNA. In the context of a large (10,000 base pair) DNA origami this can lead to lots of off target effects, leading to rapid degradation of the DNA nanostructures. The main advance of the author's work is to demonstrate a system that can have both enzyme driven logic and DNA nanostructures, by careful control of 3 nucleotide DNA sequences in the 5x5 array and choice of a specific DNA polymerase that is blocked by the TpT barrier sequence, which is then added to 3' staple ends of the DNA origami. This is an interesting and important advance in the field of switchable DNA origami structures, with potential to open up lots of future directions. The on/off switch is then used as a 'cut' and 'paste' function to build up a range of different shapes, such numbers 1-9. These results are supported by some nice AFM images of the resulting structures.

- We thank the reviewer for the appreciation of our work.

The main weakness of the work is the claims of very high yields for some of the cut and paste transitions, eg. page 7 'The corresponding yields ranged from 20% to 60%'

The data in supplementary information does not support these yield numbers. For example, SI figure 24 the 0 and 5-shaped lattices estimates yields as 30% (lanes 2,3) and 61% (lanes 5,6). The method of gel yield is to calculate the product band intensity over the total lane intensity, but in almost all gels following enzymatic reactions there is a very large and high intensity high-molecular weight smear in the gel and also clearly a lot of aggregates in the well. This includes gels in S Figs 22,24,25,26,27,28. Both the gel smear of higher molecular weight aggregates overlapping the product band and material remaining in the well will lead to an overestimate of the yield using the author's methods. Indeed, in S Fig 28 the claim is of yield 20%, when instead very obviously almost the entire sample is aggregated to the point it remains in the well.

The yield claims are also not backed up by the microscopy data, which show only 1-2 correctly formed structures for each sample and a majority of DNA is in incorrectly assembled structures or large aggregates. The authors need to address these unsupported claims of high yields before the paper can be published.

- We thank the reviewer for raising the concern. We reassessed the yields of most reactions. In the original manuscript, the yields were estimated by the ratio of target band intensity and

the intensity sum of all identifiable bands. Aggregates stuck in the gel wells and smears were not considered in the analysis. We agree with the reviewer that it leads to an overestimation of reaction yields. In the revised manuscript, the yields were estimated by the ratio of target band intensity and the intensity of the entire lane, including aggregates and smears. The recalculated yields were updated in the revised manuscript (lines 151, 153, 198, 210, 211, 218, 221; Figure 2 caption, and Supplementary Figures 4-5, 8-11 and 20-23). As for the yield analyses of cut and paste operation to produce different digit shapes, we realized that yields of complex constructs were affected by the exposure of single-stranded overhangs more than the enzymatic treatments. Misassembled lattices were produced regardless of enzymatic treatment, whose contribution to a switch reaction was not measurable easily. Therefore, we decided not to include yield analyses of this part. We added the corresponding description in the revised manuscript (lines 273- 279).

Lines 273-279:

“Exposure of single-stranded overhangs over a certain threshold was a known contributor of compromised self-assembly yield, and such a phenomenon was reported in early studies of NA nanostructures without enzymatic treatment⁴¹. A certain cut operation resulted in the creation of many single-stranded overhangs which collectively led to the random aggregation due to nonspecific base pairing. The formation yield variation of different shapes was substantial and it related more to single-strand exposure than enzymatic treatment. Therefore, yield quantification was not provided in this group of constructs.”

The following issues also need to be addressed:

1. introduction - more clearly identify what is lacking in the current state-of-the-art

- We revised the introduction section with an additional description of the challenges in the state-of-the-art practices and how we tackle the challenges in our work (lines 42-47).

Lines 42-47:

“One of the major challenges for the enzymatic treatments on DNA nanostructures is nonspecific interactions. For example, the off-target extension or cleavage can become more of an issue for DNA nanostructures with increasing complexity (MegaDalton to GigaDalton) to be treated with DNA polymerase or restriction enzymes, which could ultimately lead to byproducts such as aggregates. It is desirable to add extra layers of guarantee to further reduce nonspecific reactions³⁵.”

2. Add citations to the following key papers in enzyme-driven DNA logic circuits that involve polymerase/degradation pairs of enzymes:

Okumura, S., Gines, G., Lobato-Dauzier, N. et al. Nonlinear decision-making with enzymatic neural networks. *Nature* 610, 496–501 (2022). <https://doi.org/10.1038/s41586-022-05218-7>

Schaffter, S.W., Schulman, R. Building in vitro transcriptional regulatory networks by successively integrating multiple functional circuit modules. *Nat. Chem.* 11, 829–838 (2019). <https://doi-org.ezproxy.library.sydney.edu.au/10.1038/s41557-019-0292-z>

- We thank the reviewer for the kind suggestion. We included these key papers in the updated reference list (references 25 and 26).

3. Include references for other work that has used different methods to prevent off-target interactions of polymerases with DNA origami

Extrusion of RNA from a DNA-Origami-Based Nanofactory. Jaeseung Hahn, Leo Y.T. Chou, Rasmus S. Sørensen, Richard M. Guerra, and William M. Shih *ACS Nano* 2020 14 (2), 1550-1559 DOI: 10.1021/acsnano.9b06466

- According to the reviewer's suggestion, we described the off-target reactions in the Introduction (lines 42-47) with this paper as a reference (reference 35). In addition, we added more discussion about this point in the revised manuscript (lines 343-348).

Lines 343-348:

"In our synthetic construct design, functional sequences already serve as checkpoints to ensure the desired reaction pathways while reducing spurious reactions. It would be appealing to recruit additional components (functional DNA segments and enzymes) for a more efficient and specific switch (e.g., less off-target reactions). An inactivation of specific enzymatic activity or a total removal of a certain enzyme at a precise time point might also be useful to prevent nonspecific reactions."

4. Page2, design of switches - 'After testing several DNA polymerases and nicking endonucleases...'

- add more details on why *Bsu* DNAP and *Nt.AlwI* were selected, what other enzymes were tested and principles of selection.

- We described the rationale to adopt the enzymes in the revised manuscript (lines 69-75).

Lines 69-75:

"To satisfy our switch design for the controllable catalysis of the forward and backward reactions, we selected DNA polymerase and nicking endonuclease with special characteristics. We screened DNA polymerases with strand displacing capability to dissociate paired sticky ends or open hairpins but without exonuclease activity of undesired digestion at either the 3' or the 5' end; we also assessed nicking endonucleases whose cut sites dwell outside the recognized region to create sticky ends of diversified sequences and with a stringent recognition sequence to minimize off-target cleavage."

In fact, we assessed different DNA polymerases and nicking endonucleases to meet our screening criteria and *Bsu* DNAP and *Nt.AlwI* presented the best performance. The results of our screening were provided in the revised Supplementary Figure 1.

Supplementary Figure 1:

a DNA polymerase test

b Nickase test

Supplementary Figure 1. Enzymes screening for switch functionality. (a) DNA polymerase screening. Four DNA polymerase enzymes were tested, including *Bst* DNA polymerase large fragments, Phi 29 polymerase, *Bsu* DNA polymerase large fragments and Klenow fragment (3'→5' exo-) (NEB). Lanes L1 and L2: 1 kb+ ladder. Lane 1: structure to be

disassembled with *Bst* DNAP enzyme in 1× TE buffer. Lane 2: structure in 1× *Bst* DNAP buffer. Lane 3: structure to be disassembled with *Bst* DNAP enzyme in 1× *Bst* DNAP buffer. Lanes 4 and 10: original structures. Lane 5: structure to be disassembled with Phi 29 enzyme in 1× TE buffer. Lane 6: structure in 1× Phi 29 buffer. Lane 7: structure to be disassembled with Phi 29 enzyme in 1× Phi 29 buffer. Lane 8: structure to be disassembled with *Bsu* DNAP in 1× NEBuffer 2. Lane 9: structure to be disassembled with Klenow fragment in 1× NEBuffer 2. Disassembly results for *Bsu* DNAP and Klenow fragment were positive. We selected the *Bsu* DNAP as the working polymerase in this work. (b) Nicking endonuclease screening. We tested Nt.AlwI and Nb.BssSI to assemble two separated parts back to a coupled structure at different temperatures for 17 hours. Only Nt.AlwI produced the coupled structure under AFM imaging. Scale bar: 50 nm. Purple arrows point to the coupled structures. Pink and blue arrows point to two separated parts respectively.

5. Page 2: 'dual-thymine barriers (TpT barriers) are designed to stop the undesired extension...'
- provide more detail on mechanism for this sequence preventing further extension, this is a key feature of the later approach to DNA origami and should be emphasized with more detail earlier in the text.

- We provided more details about the working mechanism of TpT tails in the revised manuscript (lines 56-62, 132-135 and 290-292). An additional Supplementary Figure 2 about TpT barriers and TpT tails was also provided.

Lines 56-62:

“To ensure the desired enzymatic pathways and minimize side reactions for our constructs from the scaffold-free LEGO method, we designed specific restriction sites for nicking endonuclease cleavage, and three-letter coded segments for DNA polymerase extension alongside with the corresponding dual-thymine barriers (TpT barriers) to terminate extension. Since the M13 scaffold is not three-letter coded, dual-thymine tails (TpT tails) instead of TpT barriers were designed at the 3' end of common staples in DNA origami constructs to prevent undesired extension from DNA polymerase.”

Lines 132-135:

“Exonuclease activity is absent for *Bsu* DNAP, so the TpT tails remain at the 3' ends of the staples upon treatment. Such protruding overhangs would not serve as proper initiation points of *Bsu* DNAP extension, and hence the overall origami structural integrity would be preserved.”

Lines 290-292:

“Specifically, a TpT tail was appended at the 3' end of each common staple. Such an unpaired overhang remained undigested from the enzymatic treatment because of the lack of exonuclease activity for *Bsu* DNAP.”

Supplementary Figure 2:

a TpT barrier for nonworking sites

b TpT tail

Supplementary Figure 2. Schematics of TpT barrier and TpT tail working mechanism.

(a) Schematics of TpT barrier for nonworking sites. The common sticky ends with TpT barriers at both strands disabled them as template synthesized strands to disassociate the coupling partners. (b) Schematic of TpT tail. The TpT tail at the 3' end of the staple prevented *Bsu* DNAP from polymerization with the scaffold as an extension template. Tails with other sequences or lengths would function similarly. TpT was chosen because tails with consecutive dT are known with little negative impact on origami folding.

6. page 3 'complementation' doesn't make sense here, should be 'hybridization'

- We made the change accordingly (line 106).

7. page 4 'system can be cycled three rounds'

- a full cycle or round implies on-off-on, here the results are for on-off-on, so at most 1.5 cycles

- however, the gel data for this claim in the supporting figures SI Fig 10/11 is not clear at all.

Is step 1 in these gels an enzyme-driven step? Or is it just assembly of the starting material? For step 3 (lanes 3/4 in SFig 10) there is no reassembled 'X' product band visible in this gel at all, just a slightly darker smear. Similarly for step 3 in (lanes 6 in SFig 11) for the Y-design, the product band indicated by the purple arrow is barely there and not very convincing.

- In the original manuscript, we already described the challenges for the implementation of multiple rounds of switches (lines 160-163).

Lines 160-163:

“Since the forward and backward reactions proceed only with the respective enzymes, we presume that the removal of residue enzyme from the preceding reaction would be crucial for the dedicated enzyme to drive the successive reaction more efficiently, thus leading to more rounds of reversible reactions.”

To support such a hypothesis, we provided new experimental results about a compromised yield when two enzymes were added to the system on purpose in the revised manuscript (lines 163-165) and Supplementary Information (Supplementary Figure 15).

Lines 163-165:

“Such a hypothesis was supported by the experiments with both enzymes supplied on purpose (Supplementary Fig. 15).”

Supplementary Figure 15:

Supplementary Figure 15. Native polyacrylamide gel electrophoresis results of competition between *Bsu* DNAP and Nt.AlwI in the dual-pair binding system. 5-unit *Bsu* DNAP was incubated in complex X without/with dNTP, and with 2.5-unit, 5-unit, and 10-unit Nt.AlwI from left to right. Once the Nt.AlwI was added the structures worked as templates to produce short strands (at the bottom of the gel) caused by the combined effect of two enzymes, like strand displacement amplification. Thus, the disassembly of structures was discouraged when both enzymes were supplied on purpose and by-products occurred. The reaction yields (numbers in the gel) dramatically dropped to 5% with a 2:1 ratio of *Bsu* DNAP to Nt.AlwI addition. Marker: 1 kb+ ladder.

We also tuned down “three rounds of reactions” to “reversible” reactions (lines 159 and 174).

8. Generally, annotation of gels in paper and SI should more clearly indicate which samples are a result of enzyme addition, and which are DNA complexes assembled as starting materials

- We annotated the gel results in the revised Supplementary Information (Supplementary Figures 4-6, 8-11, 13, 14, 20-23, 32-37, 53, 58, and 59) according to the reviewer’s suggestion.

9. Most results show that the 'cut' operation function works much better than the 'paste' function, this needs to be addressed in the discussion

- We agree with the reviewer that the ‘cut’ operation generally performs better than the ‘paste’ operation. One of the main reasons for the performance difference is that the ‘paste’ operation is bottlenecked by the efficiency of assembly from sticky end cohesion. The enzymatic treatment is just the premise of the eventual assembly. On the other hand, the enzymatic treatments lead to the ‘cut’ operation more directly without a yield discount. However, this is not always the case. For example, the ‘cut’ yield of the FY reaction in the dual-pair system was 23%, and the ‘paste’ yield of the BY reaction in the dual-pair system was 60% (Supplementary Figure 12). Since our results did not lead to a consistent conclusion, the discussion on this point was not provided in the updated manuscript.

Supplementary Figure 12:

Supplementary Figure 12. Summary of minimalist system reaction yields. Switches X and Y in single-pair (a) and dual-pair (b) binding system. Working sites were located between two junctions outlined by purple squares (solid line: site X; dash line: site Y). Reaction efficiencies were shown in red on the corresponding reaction equations. Triplicated experiments were independently conducted (mean \pm SD, N=3). The yield of FX for the dual-pair system dropped to 63% from 91% in the one-pair system. Similarly, the yield of FY for the dual-pair system dropped to 23% from 42% in the one-pair system. The yield drop is reasonable since the overall disassembly of the complexes required would not happen till two independent decoupling events happen at the same time. On the contrary, a yield increase of the backward reaction was presented in BX and BY reactions with two working sites, 26% (dual-pair) vs 12% (one-pair) for BX and 60% (dual-pair) vs 19% (one-pair) for BY. Presumably, two neighboring sticky ends of the same construct created in the backward reactions (BX or BY) enabled the assembly in a coordinated manner, in which the initial cohesion facilitated the subsequent one due to a kinetic favor. Prolonged incubation time led to an improved efficiency of the Nt. AlwI reactions, but resulted in more aggregates for *Bsu* DNAP reactions (results not shown). Therefore, we set the incubation time as 24 hours for Nt. AlwI reactions and 2 hours for *Bsu* DNAP reactions.

10. page 4 'cooperative reactions of multiple working sites along the lattice edges can drive the overall switch reactions synergistically into completion'

This explanation makes sense for the 'paste' function, as having more possible sites to nucleate reassembly of the components would likely improve the yield. It is not clear how cooperativity could help with disassembly in the 'cut' function as well? It would be more likely to expect cooperativity to reduce the yield of 'cut' as many more bonds have to break to remove the small unit

- We agree with this reviewer that the cooperativity is more apparent in reassembly reactions. We described of enhancement more explicitly to reassembly reactions (backward reactions, BX and BY) in the revised manuscript (lines 154-160).

Lines 154-160:

“The yield increase of the backward reaction with an additional working site is consistent with the findings in our earlier study about the coordination of multiple binding pairs^{17,39}, neighboring sticky ends of the same construct created in the backward reactions enable the assembly in a coordinated manner, in which the initial cohesion facilitates the subsequent ones due to a kinetic favor. Notably, the ON/OFF switches of the dual-pair binding system were reversible (ON to OFF, back to ON, and then to OFF) with results shown in Supplementary Figs. 13 and 14.”

We also believe that there is another layer of cooperativity. Enzymes to treat a certain site have a higher probability of treating the nearby neighboring sites, which could result in a higher reaction efficiency overall. Such a substrate proximity-related favor applies to forward and backward reactions. However, the cooperativity regarding enzymatic treatments is rather

speculative without substantial support from experiments so it was not included in the revised manuscript.

Reviewer #2 (Remarks to the Author):

In this paper, the authors demonstrate the use of a nicking enzyme and polymerase action for the reconfiguration of DNA nanostructures. They first present a concept in which two DNA complexes can associate with each other and subsequently are made to dissociate. Essentially, association leads to the formation of a recognition site for a nicking enzyme. This nicking site is used as a start site for a strand-displacing DNA polymerase. Polymerization is controlled by the introduction of TT sites and the omission of dATP in the nucleotide mix.

The authors then demonstrate that a DNA lattice can be cut at predefined sites, and two lattices can be rejoined. They also show that specific shapes can be cut out from such a lattice, and that structures consisting of several connected DNA origami structures can be reconfigured.

Generally, using such enzyme-driven strand modifying operations is a very interesting concept, which is potentially useful for the realization of dynamically reconfigurable nanostructures from DNA. The overall concept is somewhat reminiscent of the “DNA toolbox” approach by Rondelez et al., which also relies on nicking enzymes and DNA polymerase strand displacement for the realization of dynamic systems.

However, it appears that in this work, the authors do not fully realize the potential of the approach, and the progress made is not as significant as one might expect. Specifically, it is not clear how programmable this approach is in the end, how quickly the processes occur (compared to, e.g., ordinary strand displacement), the robustness of the procedures/their yield, and ultimately, whether there is any significant advantage over existing or alternative methods. These concerns should be addressed in a revised version of the manuscript.

- We thank the reviewer for the constructive criticism. According to the suggestion, we have added new experiments for comparison between our enzyme-mediated systems and toehold-mediated systems. In the side-by-side comparison, we can see clear advantages in reaction yield and/or rate for our enzyme-mediate systems (Supplementary Figures 16 and 17). Moreover, the components are simpler (no invading strands necessary) and free from stoichiometry adjustment. We described the comparison in the revised manuscript (lines 179-189, and 336-339).

Lines 179-189:

“Notably, when compared to the toehold-mediated strand displacement system, apparent advantages were presented in our enzyme-mediated system. In FX reactions, although yields from the enzyme-mediated system (46%) and the toehold-mediated one (47%) were similar after a 2-hour incubation(Supplementary Fig. 16a), the reaction rate of the enzyme-mediated system was much higher than that of the toehold-mediated system. As shown in the time course results (Supplementary Fig. 16b), the enzymatic reaction reached an equilibrium in less than 3 minutes (reaction rate at 3 min^{-1}), while it took more than 90 minutes to reach a similar level of equilibrium in control experiments of toehold-mediated strand displacement (reaction rate at 0.03 min^{-1}). In BX reactions (Supplementary Fig. 17), the assembly yield of the enzyme-mediated system was measured at about 29%, but no obvious assembled product was generated in our control of the toehold-mediated system.”

Lines 336-339:

“According to the side-by-side comparison with the control system based on toehold-mediated strand displacement, the reaction yield and/or rate of our enzyme-mediated system was superior (Supplementary Fig. 16), not to mention the simpler components and independence of stoichiometry.”

Supplementary Figure 16:

Supplementary 16. Comparison of the enzyme-mediated system and toehold-mediated strand displacement system in the FX reaction of the dual-pair binding system. In the enzyme-mediated reaction, the complex X, at a concentration of 250 nM, was mixed with 5 U *Bsu* DNAP for incubation at 37°C for 2 hours. In a toehold-mediated reaction (a), the complex X with the toehold, also at a concentration of 250 nM, was mixed with 250 nM toehold invader strands for incubation at 37°C for 2 hours. (b) Native polyacrylamide gel electrophoresis results of the time course for *Bsu* DNAP enzyme and toehold invader strand. L: 1 kb+ ladder. (c) Time-dependent FX reaction in *Bsu* DNAP enzyme (red dots) and toehold invader strand displacement (blue dots), resolved by native polyacrylamide gel electrophoresis. The percentage of X_I was quantified by an exponential fit. The reaction rate constants are 3 min⁻¹ (± 0.65) and 0.03 min⁻¹ (± 0.004) for enzyme- and toehold-mediated systems respectively, extracted from curves. Triplicated experiments were independently conducted (mean ± SD, N=3).

Supplementary Figure 17:

Supplementary Figure 17. Comparison of the enzyme-mediated system and toehold-mediated strand displacement system in the BX reaction of the dual-pair binding system.

In the enzyme-mediated reaction, the X_I and X_{II} , at a concentration of 750 nM, were mixed with 15 U Nt.AlwI enzyme for an incubation at 37°C overnight. In a toehold-mediated reaction (a), the X_I and X_{II} with toehold, also at a concentration of 750 nM, were mixed with 750 nM toehold invader strands for incubation at 37°C overnight. (b) Native polyacrylamide gel electrophoresis results of the dual-pair binding system BX reaction. Lane L: 1 kb+ ladder. Lane 1: dual-pair binding site unit X_I . Lane 2: dual-pair binding site unit X_{II} . Lane 3: the formation of complex X (indicated by the purple triangle, yield ~29%) from X_I and X_{II} upon Nt.AlwI supplied. Lane 4: dual-pair binding site unit X_I with toehold. Lane 5: dual-pair binding site unit X_{II} with toehold. Lane 6: the formation failure of complex X from X_I and X_{II} with toehold upon anti-invader strands supplied.

We also emphasized off-target extension and cleavage as a general challenge for the enzymatic treatments on DNA nanostructures. According to our design of reaction cycles, specific pathways are guaranteed by the enzyme choice and sequence features of DNA constructs. We described the corresponding claims in the revised manuscript (lines 343-346).

Lines 343-346:

“In our synthetic construct design, functional sequences already serve as checkpoints to ensure the desired reaction pathways while reducing spurious reactions. It would be appealing to recruit additional components (functional DNA segments and enzymes) for a more efficient and specific switch (e.g., less off-target reactions).”

Specific points:

- What is the “Yin and Yang theory”? This reviewer believes that Yin and Yang are not scientific concepts and are not well-defined. This should be phrased more carefully, e.g., “similar to Yin and Yang, the reactions complement each other, etc...”. It is notable that the term “Yin and Yang” only appears in the title of the cited reference 23, likely because it sounds appealing, but not more.

- We removed the Yin and Yang analogy from the main text. However, we couldn't find a proper alternative to describe the opposite but complementary outcome of the enzyme pair, so we decided to keep the Taiji diagram to depict the cycling of assembly and disassembly.

- In the reaction cycle in Fig. 1b, it is unclear to the reviewer whether switch Y would predominantly be present in the extended form (open hairpin) on top or if it would rapidly interconvert between the hairpin form (on the right) and other intermediates. Perhaps the hairpin is even more thermodynamically stable than the extended form?

- We thank the reviewer to point it out. We provided the thermodynamic analysis of complex Y. With three entities (y_{H-yc^*} , y_{E^*} , and y) as input, simulated results from NUPACK show the predominant product as the coupled complex rather than the closed hairpin. The thermodynamic analysis was added in the Supplementary Information (Supplementary Figure 3).

Supplementary Figure 3:

Supplementary Figure 3. Thermodynamic analysis of ON state in Fig. 2b. (a) Schematic of complex Y in switch Y. (b) NUPACK^[1] analysis of complex Y. With three entities (y_{H-yc^*} , y_{C^*} , y_{E^*} , and y) as the input, simulated results from NUPACK showed the predominant product as the coupled complex rather than the closed hairpin.

- The reviewer does not understand why one of the states is labeled “ON” and the other “OFF”.

- We thank the reviewer for pointing out the confusion. We specified ON as the assembled/coupled state and OFF as the disassembled/decoupled state (line 55).

- The overall yields of the reactions are not particularly impressive. Given this, what is the advantage of a procedure that requires the addition of enzymes, resulting only in partial reconfiguration? Multiple cycles will further reduce the overall yield.

Maybe one can highlight and discuss the differences to other approaches for reconfigurability, and why this one is superior. If it is not, what has to be developed?

- As we mentioned above, we added the new experiments of the control system of toehold-mediated strand displacement. According to the side-by-side comparison, our enzyme-mediated system was shown with superior reaction yield and/or rate than the toehold-mediated system.

We agree with the reviewer that the general yields and reversibility of our reaction systems are not super. We plan to further improve the systems with engineered enzymes for more specific treatments, and blocking reagents (e.g., BSA and SSB) to prevent single-strand DNA from nonspecific interactions.

- While the concept in Fig. 1 promises the ability to switch between two different designs, this is only demonstrated once with DNA nanostructures (Fig. 2c and 2f). Figures 3 and 4, on the other hand, are solely based on the polymerase-driven FX and BY (which might be more efficient?). This suggests that the potential strengths of the concept aren't fully realized in its current form.

- We agree with this reviewer about the emerging research opportunities, including the one about using nicking endonuclease to realize composite cut and paste operations. Polymerase-based reactions led to higher yields, so we focused on the corresponding composite reactions. Such a rationale is provided in the revised manuscript (lines 233-235).

Lines 233-235:

“It is also feasible to design a composite operation based on nicking endonuclease (Supplementary Fig. 28). Due to the higher reaction yields (FX>FY, BY>BX, Supplementary Fig. 12), we focused on DNA polymerase for the composite operation in this work.”

A similar composite cut and paste operation can be executed with nicking endonuclease and the schematics were presented in Supplementary Figure 28. We focused on the polymerase based reaction in order to showcase the concept of composite operation. Therefore, a more comprehensive demonstration is not provided in this work.

Supplementary Figure 28:

Supplementary Figure 28. Schematics one-step cut and paste on 8×8 lattice by Nt.AlwI nicking endonuclease. (a) Schematic of cut operation with nicking endonuclease (e.g., FY reaction) to create a C-shape lattice. (b) Schematic of paste operation with nicking endonuclease (e.g., BX reaction) to form an 8×16 lattice. (c) Schematic of the composite cut (FY reaction) and paste (BX reaction) operation with nicking endonuclease to rearrange a 0-shaped lattice. The cropped lattices are different from the system with *Bsu* DNAP.

- Cutting and rejoining reactions are not freely programmable. Essentially, the break/cutting points must be designed into the devices. Also there are sequence constraints due to the use of a 3-letter code. Developing more general reconfigurability would be intriguing.

- The TpT tail is a general design and it is applicable to all the common staples of our origami constructs. According to our cut and paste designs for 8×8 lattice and origami square, we can place the 3-letter coded segment to be enzymatically treated in arbitrary edges/boundaries. We suppose our design scheme to be programmable. We agree with the reviewer that it is not freely programmable and we proposed a more arbitrary cutting with Cas9 instead of nicking endonuclease in the revised manuscript (lines 283-284). However, that falls out of the main scope of this study, and the experimental implementations are not included.

Lines 283-284:

“Furthermore, one could also use Cas9 to substitute nicking endonuclease to achieve a more arbitrary cleavage without relying on predefined working sites.”

- In Fig. 4, the authors use the term 'reconfiguration'. It should be clarified whether the structures individually undergo internal reconfiguration (possibly by linking the substructures together) or if they all merely disassemble and reassemble in a different form.

- We thank the reviewer to point out the ambiguity about terminology. Since the process might still be a combination of disassembly and reassembly, we decided not to use the term 'reconfiguration' to describe the shape change. We modified the terminology in the revised manuscript (lines 305, 307, 312 and 318).

Reviewer #3 (Remarks to the Author):

Nice work! Kang and Wei have developed a DNA switch system driven by a combination of DNA restriction nicking enzyme and DNA polymerase with strand displacement capability. The two enzymes work in the opposite directions: one enzyme works to associate two DNA molecules and the other works to dissociate the two DNA molecules. The functions of the two enzymes in terms of association-dissociation depends on the specific design; but in either case, they are specific and controllable. The authors have applied this strategy to multiple systems of DNA nanostructures. Each system has been concretely demonstrated by native gel electrophoresis and atomic force microscopy. Excellent idea and concrete experimental demonstration! This work is a great addition to the tool box of regulating DNA self-assembly. This referee would strongly recommend for publication.

- We thank the reviewer for the appreciation of our work.

REVIEWERS' COMMENTS

Reviewer #1 (Remarks to the Author):

The authors have addressed all the review comments appropriately and I recommend the manuscript for publication.

The comparison of the enzymatic and toehold strand displacement systems and the addition of screening data for the different polymerases and nicking enzymes were particularly helpful. These results will benefit many researchers, allowing them to build on this work.

Reviewer #2 (Remarks to the Author):

The authors addressed all of this reviewer's concerns and improved the manuscript accordingly.

This reviewer has one more question, maybe the reviewer overlooked it in the manuscript:
Are the enzyme reactions somehow terminated? For instance, after addition of the polymerase (FX reaction) is the DNAP deactivated and then the nicking enzyme added? Are the solutions somehow purified between the steps?

REVIEWERS' COMMENTS

Reviewer #1 (Remarks to the Author):

The authors have addressed all the review comments appropriately and I recommend the manuscript for publication.

The comparison of the enzymatic and toehold strand displacement systems and the addition of screening data for the different polymerases and nicking enzymes were particularly helpful. These results will benefit many researchers, allowing them to build on this work.

We thank the review for the kind appreciation of our work.

Reviewer #2 (Remarks to the Author):

The authors addressed all of this reviewer's concerns and improved the manuscript accordingly.

This reviewer has one more question, maybe the reviewer overlooked it in the manuscript: Are the enzyme reactions somehow terminated? For instance, after addition of the polymerase (FX reaction) is the DNAP deactivated and then the nicking enzyme added? Are the solutions somehow purified between the steps?

We thank the review for the kind appreciation of our work. According to the reviewer's suggestion, we specified the experimental details as "However, the thorough inactivation or removal of a chosen enzyme after its designated reaction was not carried out due to practical difficulties." (Page 5 lines 172-173)